# Oral Contraceptive Use and Reproductive History in Relation to Metabolic Syndrome Among Women from KNHANES 2010–2023

**DOI:** 10.3390/jcm14176319

**Published:** 2025-09-07

**Authors:** In Ae Cho, Jaeyoon Jo, Jeesun Lee, Hyunjin Lim, Yun-Hong Cheon, Rock Bum Kim

**Affiliations:** 1Institute of Health Sciences, Gyeongsang National University College of Medicine, Jinju 52727, Republic of Korea; obgychoia@gnu.ac.kr (I.A.C.); poodeeng@gnu.ac.kr (J.J.); 2Department of Obstetrics and Gynecology, Gyeongsang National University Hospital, Jinju 52727, Republic of Korea; grazysun@gnuh.co.kr; 3Division of Rheumatology, Department of Internal Medicine, Gyeongsang National University Hospital, Jinju 52727, Republic of Korea; lhj8905@gnu.ac.kr; 4Regional Cardiocerebrovascular Disease Center, Gyeongsang National University Hospital, Jinju 52727, Republic of Korea

**Keywords:** metabolic syndrome, oral contraceptives, reproductive factors, menopausal status

## Abstract

**Background/Objectives**: This study examined how reproductive factors—such as oral contraceptive (OC) use, age at menarche, number of pregnancies, and age at first delivery—are related to the risk of metabolic syndrome (MetS) in Korean women aged 30–69, based on their menopausal status. **Methods**: Data from the Korea National Health and Nutrition Examination Survey 2010–2023 were analyzed, including 31,178 women with complete data. Survey-weighted logistic regression and restricted cubic spline analyses were conducted, adjusting for sociodemographic, lifestyle, and reproductive covariates. **Results:** OC use was associated with higher MetS risk in both pre-menopausal (adjusted OR 1.40, 95% CI 1.13–1.72) and post-menopausal women (adjusted OR 1.16, 95% CI 1.03–1.29). This association was observed primarily in relation to elevated blood pressure in both groups and high triglycerides in post-menopausal women. Other reproductive factors, including age at menarche, number of pregnancies, and age at first delivery, showed no significant associations with MetS risk. **Conclusions**: OC use was associated with higher MetS risk in this cross-sectional study of Korean women. These observational findings suggest a potential relationship that warrants further investigation through longitudinal studies to establish temporal relationships and explore underlying mechanisms.

## 1. Introduction

Metabolic syndrome (MetS) is a cluster of metabolic abnormalities, including abdominal obesity, hypertension, dyslipidemia, and impaired glucose metabolism. These factors increase the risk of cardiovascular disease, type 2 diabetes, and all-cause mortality [1]. Well-established risk factors for MetS include older age, higher body mass index (BMI), low physical activity, and unhealthy lifestyle behaviors [2]. In women, unique physiological states such as puberty, pregnancy, and menopause, as well as reproductive factors like age at menarche, parity, and age at first delivery, are increasingly recognized as key determinants of MetS and other chronic diseases, including diabetes and cardiovascular disease [3,4,5,6]. Hormonal and metabolic changes throughout a woman’s reproductive life may influence her long-term metabolic health.

Many studies have explored the relationship between female reproductive factors and MetS, but findings remain inconsistent [7,8,9,10,11,12]. Some research has shown that early menarche is associated with a higher risk of MetS and related disorders, while other studies have found a higher risk with both early and late menarche, suggesting a reversed J-shaped association [7,9,10]. Similarly, large population-based studies reported associations between age at menarche, number of pregnancies, age at first birth, and MetS risk, though results varied by population and analytic method [13]. These inconsistencies highlight the need for further studies that use large representative samples and stratify by menopausal status.

Compared with reproductive history, the link between oral contraceptive (OC) use and MetS has received less attention, despite widespread OC use among women of reproductive age. Prior studies suggest that OC use may worsen metabolic markers such as blood pressure and lipid levels, but findings on its association with MetS are limited and mixed [14,15,16]. Recent lipidomic and hemodynamic studies show that modern OC formulations may affect vascular resistance, insulin sensitivity, and lipid metabolism, potentially increasing cardiometabolic risk [17,18]. Given the potential metabolic effects of hormonal contraceptives, further research is needed across diverse populations and reproductive stages.

This study aimed to examine the associations between OC use and reproductive factors—including age at menarche, number of pregnancies, age at first delivery, and breastfeeding history—with the risk of MetS among Korean women. We used data from a large, nationally representative survey and conducted analyses stratified by menopausal status to address a key limitation of prior research and explore differences pre and post menopause.

## 2. Materials and Methods

### 2.1. Data Source and Study Population

This study utilized data from the Korea National Health and Nutrition Examination Survey (KNHANES), an ongoing, nationally representative surveillance program conducted by the Korea Disease Control and Prevention Agency. KNHANES assesses the health and nutritional status of the Korean population and monitors trends in health-related risk factors and the prevalence of major chronic diseases. The survey employs a complex, multistage probability sampling design to ensure representativeness for the civilian, non-institutionalized Korean population. Detailed information on the survey design, sampling methods, and data collection procedures is available on the KNHANES website (http://knhanes.kdca.go.kr/ (accessed on 1 August 2025)).

Data from the 2010–2023 KNHANES cycles were combined based on the official analytic guidelines. Combined survey weights were calculated and applied to all analyses. These weights accounted for the complex sampling design and year-to-year integration. Among women who participated in the KNHANES from 2010 to 2023, those aged 30 to 69 years were initially selected for this study. Participants were excluded if they had incomplete or implausible data for key reproductive variables, such as age at menarche, menopause, number of pregnancies, age at first delivery, breastfeeding duration, or history of OC use. After these exclusions, 31,178 women with complete data remained for the final analysis. These participants were further classified into pre-menopausal (*n* = 16,323) and post-menopausal (*n* = 14,857) groups based on their menopausal status (Figure 1). Menopausal status was determined using the survey question “Are you currently having menstruation?” with the response options “natural menopause” or “artificial menopause”. Women who selected either option and additionally reported their age at menopause were classified as post-menopausal. Those who reported ongoing menstruation were considered pre-menopausal. The questionnaire did not collect further details regarding the reasons for artificial menopause (e.g., bilateral oophorectomy, hysterectomy, or other causes). Information on hormone replacement therapy use was not available in the dataset and thus could not be considered in the classification.

### 2.2. Covariates for Adjustment

Sociodemographic, lifestyle, and reproductive variables were included as covariates from the KNHANES. Body mass index (BMI) was calculated as weight in kilograms divided by the square of height in meters (kg/m^2^), using measured values of height and weight. Education level was classified as ≤elementary school, middle school, high school, and ≥college. Monthly household income was divided into quintiles (Q1: low, Q2: low–middle, Q3: middle, Q4: middle–high, Q5: high). Marital status was defined as having a spouse or not. Smoking status was categorized as never smoker, ex-smoker, or current smoker. Drinking habits were classified as never drinking, normal drinking (less than or equal to 4 times per month), or high drinking (more than 4 times per month). Physical activity was assessed by the number of days per week participants walked for at least 10 min (0, 1–3, or 4–7 days) and the number of days per week they performed strength exercises (0, 1–3, or 4–7 days).

Reproductive factors included age at menarche (years), number of pregnancies, age at first delivery (years), and number of children breastfed. Menopausal status (post-menopausal or pre-menopausal) and history of OC use (yes or no) were also included as covariates. OC use was defined based on a standardized question (‘Have you ever taken oral contraceptive pills for more than one month?’). Those who answered ‘yes’ were classified as OC users. All covariates were measured using standardized questionnaires and protocols as part of the KNHANES. Covariates with missing values were not imputed and were analyzed as observed. The proportion of missing data for each covariate is provided in Appendix A.

### 2.3. Metabolic Syndrome as the Study Outcome

MetS, the primary outcome of this study, was defined according to the criteria established by the National Cholesterol Education Program Adult Treatment Panel III. MetS was diagnosed when three or more of the following five components were present: (1) increased waist circumference (>88 cm for women); (2) elevated triglycerides (≥150 mg/dL) or current use of lipid-lowering medication; (3) reduced high-density lipoprotein (HDL) cholesterol (<40 mg/dL for men and <50 mg/dL for women) or current use of medication for reduced HDL cholesterol; (4) elevated blood pressure (≥130/85 mmHg) or current use of antihypertensive medication; and (5) elevated fasting glucose (≥100 mg/dL) or current use of antidiabetic medication. This definition does not require any single component as mandatory; the presence of any three or more of the five criteria qualifies as MetS.

### 2.4. Statistical Analysis

All statistical analyses were conducted using R software (version 4.4.3), incorporating the complex survey design and sampling weights of the KNHANES. The survey design was specified using the svydesign function of the survey package (version 4.4-2), which accounted for primary sampling units, stratification, and sampling weights to ensure nationally representative estimates. For pooled analyses across 2010–2023, survey weights were recalculated following official KNHANES guidance. Each participant’s weight was adjusted by dividing the original weight by the total number of combined survey years (14 years). This approach ensures that the integrated weights correspond to the national population for the pooled period, as recommended in the KNHANES guidelines [19].

Descriptive statistics were calculated for all covariates according to the presence or absence of MetS. For categorical variables, frequencies and percentages are presented, and, for continuous variables, means and standard deviations are reported. Both unweighted descriptive statistics for the number of participants and survey-weighted descriptive statistics were generated. Standardized mean differences (SMDs) were calculated to assess the balance of covariates between groups.

To examine the association between covariates and MetS, survey-weighted logistic regression models were used. Univariable logistic regression analyses were first performed for each covariate to estimate the crude odds ratios (cORs) and 95% confidence intervals (CIs) for MetS. Subsequently, multivariable logistic regression models were constructed, including all covariates simultaneously, to estimate adjusted ORs (aORs) and 95% CIs. All analyses incorporated survey weights to account for the complex sampling design.

To evaluate the risk of MetS according to reproductive factors, restricted cubic spline (RCS) analyses were performed separately for pre-menopausal and post-menopausal women. The RCS method flexibly models nonlinear associations between reproductive variables and MetS. These variables include age at menarche, number of pregnancies, and age at first delivery. We used four knots placed at the 5th, 35th, 65th, and 95th percentiles of each exposure distribution, based on default quantile placement. Goodness-of-fit for each RCS model was assessed using Nagelkerke’s pseudo R^2^ and the C-statistic, and these values are presented on the corresponding spline plots. All covariates were adjusted for in the models. The results were visualized with the aOR and 95% CI plotted against the reproductive factors. Separate plots were generated for pre-menopausal and post-menopausal groups to assess potential differences in associations by menopausal status.

In addition to examining MetS defined by three or more risk components, we also evaluated OC use based on stricter thresholds. These included the presence of all five, four or more, two or more, or at least one component. Furthermore, we examined the association between OC use and each individual component of MetS, namely high blood pressure, elevated fasting blood glucose, high triglycerides, low high-density lipoprotein (HDL) cholesterol, and abdominal obesity (excessive waist circumference). These analyses were conducted separately for pre-menopausal and post-menopausal women, and aORs with the 95% CI were estimated for each outcome. We also analyzed the results using a waist circumference cutoff of 80 cm for MetS diagnosis.

For sensitivity analysis, we examined the association between OC use duration and MetS using data from the 2010–2012 survey cycle, which collected information on OC use duration. Multivariable survey-weighted logistic regression models were stratified by menopausal status, and adjusted odds ratios with 95% CIs were estimated for categorized durations of OC use (never used, less than 1 year, 1–3 years, 3–5 years, and 5 years or more).

The full R code used for all analyses in this study is provided in the Appendix A for reproducibility.

## 3. Results

### 3.1. Baseline Characteristics by Metabolic Syndrome

Table 1 presents the baseline characteristics of the study population according to the presence or absence of MetS. Among the 31,178 women included, 8324 (26.7%) had MetS. Women with MetS were older (mean age: 57.4 vs. 47.4 years; SMD = 0.993) and had a higher BMI (26.1 vs. 22.7 kg/m^2^; SMD = 1.046) compared with those without MetS. The MetS group also had a higher proportion of overweight or obese individuals (BMI ≥ 25.0 kg/m^2^; SMD = 0.967).

In addition, women with MetS had a later age at menarche (14.7 vs. 13.9 years; SMD = 0.376), more pregnancies (3.74 vs. 2.94; SMD = 0.432), a younger age at first delivery (24.73 vs. 26.64 years; SMD = 0.459), and more breastfed children (1.90 vs. 1.05; SMD = 0.430). Post-menopausal status was more common in the MetS group (74.3% vs. 37.7%; SMD = 0.775), as was OC use (21.1% vs. 13.5%; SMD = 0.180). Other lifestyle factors, such as household income (SMD = 0.162), smoking status (SMD = 0.075), walking days per week (SMD = 0.085), and strength exercise days per week (SMD = 0.165), showed smaller differences between groups.

### 3.2. Associations Between Covariates and Metabolic Syndrome by Menopausal Status

Table 2 shows the associations between covariates and the risk of MetS from the multivariable analysis. The crude ORs are shown in Appendix A. In both pre-menopausal and post-menopausal women, higher age and BMI were strongly associated with increased risk of MetS. For each 1-year increase in age, the aOR for MetS was 1.11 (95% CI: 1.10–1.12) in pre-menopausal women and 1.09 (95% CI: 1.08–1.10) in post-menopausal women. Each 1 kg/m^2^ increase in BMI was associated with an aOR of 1.45 (95% CI: 1.42–1.48) and 1.31 (95% CI: 1.29–1.33) in pre- and post-menopausal women, respectively.

Lower income was also linked to a higher MetS risk. Compared with the highest-income group, the lowest quintile had an aOR of 1.45 (95% CI: 1.14–1.84) in pre-menopausal women and 1.27 (95% CI: 1.10–1.48) in post-menopausal women. Current smoker was associated with higher MetS risk in both groups (aOR: 1.42 (95% CI: 1.03–1.96) for pre-menopausal women; 1.60 (95% CI: 1.26–2.04) for post-menopausal women), as was high drinking (aOR: 2.13 (95% CI: 1.58–2.88) and 1.41 (95% CI: 1.16–1.73), respectively).

Engaging in strength exercise was inversely associated with MetS in post-menopausal women (aOR for 1–3 or 4–7 days/week: 0.62 (95% CI: 0.52–0.75)), but not in pre-menopausal women. OC use was associated with increased MetS risk in both groups (aOR: 1.40 (95% CI: 1.13–1.72) for pre-menopausal women; 1.16 (95% CI: 1.03–1.29) for post-menopausal women). After adjustment, age at menarche, number of pregnancies, age at first delivery, and number of breastfed children were not significantly associated with MetS in either group.

### 3.3. Associations Between Reproductive Factors and Metabolic Syndrome

Figure 2 illustrates the results of restricted cubic spline analyses assessing potential nonlinear relationships between reproductive factors and MetS risk, stratified by menopausal status. For age at menarche, no significant nonlinear association with MetS risk was found in either pre-menopausal (*p* for overall = 0.929) or post-menopausal women (*p* for overall = 0.655). The aORs remained stable across the age range of 8–23 years, with confidence intervals including the null value of 1.0 in both groups.

Similarly, number of pregnancies showed no significant nonlinear association with MetS risk in either pre-menopausal (*p* for overall = 0.364) or post-menopausal women (*p* for overall = 0.460). While a slight downward trend in MetS risk was observed with an increasing number of pregnancies among pre-menopausal women, it was not statistically significant. In post-menopausal women, the association remained flat across the full pregnancy range (0–13).

Age at first delivery also showed no significant nonlinear relationship with MetS risk in either group (*p* for overall = 0.118 for pre-menopausal women; 0.172 for post-menopausal women), with aORs remaining close to 1.0 throughout the age range of 15–43 years.

### 3.4. Association Between Oral Contraceptive Use and Metabolic Syndrome

Figure 3 presents the association between OC use and MetS risk in both pre-menopausal and post-menopausal women, considering different definitions of MetS and its individual components. OC use was significantly associated with increased risk of MetS—defined as having three or more metabolic risk factors—in both pre-menopausal (aOR: 1.40, 95% CI: 1.13–1.72) and post-menopausal women (aOR: 1.16, 95% CI: 1.03–1.29).

However, no significant associations were found when stricter (≥4 components or all 5 components) or more lenient (≥2 components or ≥1 component) definitions of MetS were applied. When examining individual MetS components, OC use was associated with higher risk of high blood pressure in both pre-menopausal (aOR: 1.30, 95% CI: 1.17–1.45) and post-menopausal women (aOR: 1.20, 95% CI: 1.01–1.43), and with high triglycerides in post-menopausal women (aOR: 1.23, 95% CI: 1.04–1.44). No significant associations were observed for elevated fasting glucose, low HDL cholesterol, or increased waist circumference in either group.

### 3.5. Sensitivity Analysis

Table 3 presents the results of the sensitivity analysis. Among pre-menopausal women, no statistically significant associations were observed across any duration categories, with adjusted ORs ranging from 0.44 (95% CI: 0.13–1.49) for ≥5 years of use to 1.50 (95% CI: 0.06–40.62) for 3–5 years of use. In post-menopausal women, while shorter durations of OC use showed no significant associations, long-term use (≥5 years) was significantly associated with increased odds of MetS (aOR: 2.13, 95% CI: 1.27–3.57).

To examine the sensitivity of our findings regarding the relationship between OC use and MetS to different diagnostic thresholds, we conducted supplementary analyses using a waist circumference cutoff of 80 cm for MetS diagnosis. These results are presented in Appendix A.

## 4. Discussion

In this large, nationally representative cross-sectional study of Korean women aged 30–69 years, we assessed the associations between reproductive factors—including OC use—and the risk of MetS, stratified by menopausal status. The key findings are as follows: (1) OC use was significantly associated with increased risk of MetS in both pre- and post-menopausal women; (2) other reproductive factors (e.g., age at menarche, number of pregnancies, age at first delivery, breastfeeding) were not independently associated with MetS after adjustment; and (3) OC use was particularly associated with elevated blood pressure and triglyceride levels.

Unlike some previous studies, we did not observe significant associations between reproductive factors and MetS after multivariable adjustment. Kwan et al. [7] and Zuo et al. [13] found a J-shaped association between age at menarche and MetS, while a study by Ling et al. [12] reported no association, consistent with our results. These discrepancies may reflect methodological differences. Many earlier studies did not stratify by menopausal status, which is a key modifier of metabolic risk [20,21]. Additionally, population-specific factors such as genetics, diet, or cultural norms may affect these associations [22]. The lack of associations may also result from collinearity with age or BMI, both of which are strong predictors of MetS [10]. However, in this study, variance inflation factors were <2.0, suggesting no serious multicollinearity. Another possibility is that improvements in maternal and women’s healthcare over recent decades have diminished the predictive value of these reproductive markers [23]. Taken together, these considerations highlight the importance of accounting for the population context and methodological rigor in interpreting epidemiological findings related to women’s reproductive health and long-term metabolic outcomes.

OC use was consistently associated with MetS in both pre-menopausal and post-menopausal women, with adjusted odds ratios of 1.40 and 1.16, respectively. Sensitivity analysis by OC duration categories revealed that, among post-menopausal women, ≧5 years of OC use was associated with MetS (adjusted OR = 2.13), while no significant duration-related associations were observed in pre-menopausal women. This association appeared to be primarily driven by elevations in blood pressure in both groups and by hypertriglyceridemia specifically in post-menopausal women. These findings are supported by several biological mechanisms proposed in the literature [24,25,26]. Estrogen, particularly ethinylestradiol, can impair lipid profiles by raising triglycerides and reducing LDL clearance [27,28]. Some androgenic progestins, such as levonorgestrel, may also increase blood pressure through sodium retention [25,29]. Moreover, a “metabolic memory” effect may persist in post-menopausal women after early OC exposure [30].

While our findings suggest a modest but consistent association between OC use and MetS risk, the literature offers mixed evidence. A recent 2023 cohort study using UK Biobank found no increased cardiovascular disease or mortality risk from OC use and even suggested protective effects among long-term users [31]. However, that study focused on different outcomes and included mainly White British women, limiting relevance to our Korean population.

In our analysis, OC use was not associated with stricter definitions of MetS or with glucose and waist circumference. This suggests that OC use may selectively affect certain metabolic domains, rather than causing global dysfunction. These findings highlight the importance of incorporating reproductive and hormonal history into metabolic risk assessment. The 2024 U.S. Medical Eligibility Criteria for Contraceptive Use acknowledge these metabolic considerations, recommending caution with combined OCs in women with existing cardiometabolic risk factors, particularly those with hypertension or diabetes [32].

Finally, strength training was inversely associated with MetS only among post-menopausal women. This may reflect the greater metabolic benefits of resistance training in older women, who are more susceptible to sarcopenia and visceral fat accumulation after menopause [33]. These findings suggest that incorporating strength-based interventions could be particularly effective in mitigating metabolic risk in this population.

The strengths of this study include the large sample size, nationally representative design, detailed reproductive and metabolic measurements, and stratified analytic approach by menopausal status. The use of complex survey weights and advanced statistical modeling enhances the robustness and generalizability of our findings. Our restricted cubic spline analyses further confirmed the lack of significant nonlinear associations between reproductive factors and MetS, regardless of menopausal status.

Despite these strengths, several limitations should be considered. First, the cross-sectional design limits causal inference as exposure and outcome were assessed simultaneously, preventing the establishment of temporal relationships. Reverse causality is possible, as women with metabolic risk factors may have avoided or discontinued OC use. Our findings should therefore be interpreted as associations rather than causal links. Second, self-reported reproductive histories may introduce recall bias, particularly among older post-menopausal women, potentially causing exposure misclassification. To mitigate this bias, we excluded individuals with incomplete reproductive data to enhance validity. Third, our data lacked detailed OC characteristics (duration, formulation, initiation age), limiting nuanced analysis. Most Korean OCs during this period were low-dose combined pills (20–30 µg ethinyl estradiol), so findings likely reflect second- and third-generation OC effects. Fourth, despite extensive covariate adjustment, residual confounding from unmeasured dietary, genetic, and psychosocial factors remains possible. Accordingly, interpretation of the results should consider this potential limitation.

## 5. Conclusions

This cross-sectional study found that OC use was associated with increased MetS risk among Korean women, regardless of menopausal status, mainly through its associations with blood pressure and triglyceride levels. Other reproductive factors showed no significant associations with MetS after covariate adjustment. While these observational findings suggest potential relationships between hormonal contraceptive use and metabolic outcomes, longitudinal studies are needed to establish temporal relationships and better understand potential causal pathways. These results may inform clinical discussions about contraceptive choices and metabolic monitoring in women with existing cardiometabolic risk factors.

## Figures and Tables

**Figure 1 jcm-14-06319-f001:**
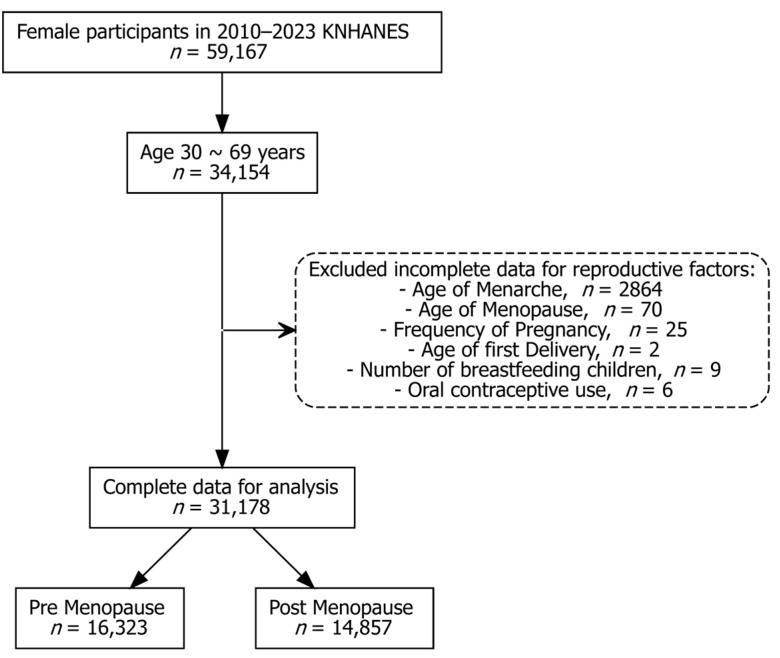
Flowchart for participant selection.

**Figure 2 jcm-14-06319-f002:**
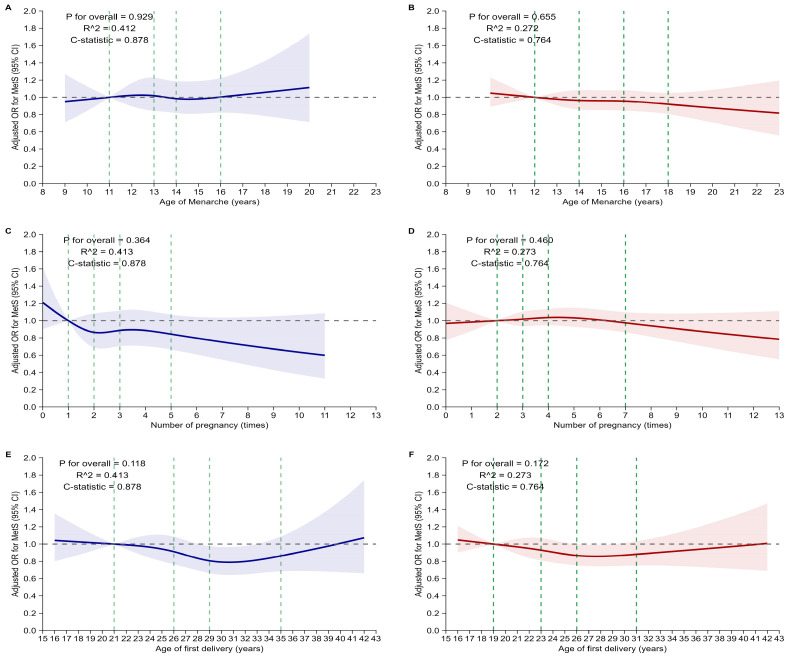
Restricted cubic spline analyses of reproductive factors and risk of metabolic syndrome stratified by menopausal status. Restricted cubic spline curves illustrate the adjusted ORs and 95% confidence intervals (shaded areas) for MetS according to age at menarche (**A**,**B**), number of pregnancies (**C**,**D**), and age at first delivery (**E**,**F**), separately for pre-menopausal (blue, left panels) and post-menopausal women (red, right panels). Models were adjusted for age, body mass index, number of pregnancies, age at first delivery, number of breastfed children, OC use, education level, income quintile, marital status, smoking status, drinking habits, walking frequency, and exercise frequency. Survey-weighted estimates were adjusted for all covariates. Knots were placed at the 5th, 35th, 65th, and 95th percentiles of each exposure distribution and are indicated by green dashed lines. Nagelkerke’s R^2^ and C-statistics are shown in each panel to indicate model performance. Abbreviations: MetS, metabolic syndrome; OC, oral contraceptive.

**Figure 3 jcm-14-06319-f003:**
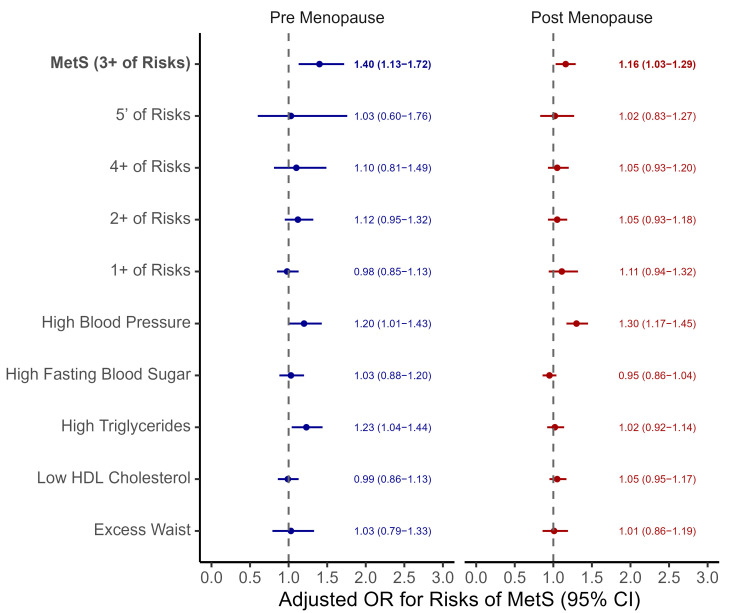
Adjusted odds ratios for metabolic syndrome and its components according to OC use, stratified by menopausal status. Forest plots display adjusted odds ratios (ORs) and 95% confidence intervals (CIs) for the association between OC use and the risk of MetS and its individual components in pre-menopausal (left) and post-menopausal (right) women. All models were adjusted for age, body mass index, reproductive factors, education, income, marital status, smoking, alcohol consumption, and physical activity. Abbreviations: HDL, high-density lipoprotein; MetS, metabolic syndrome; OC, oral contraceptive.

**Table 1 jcm-14-06319-t001:** Baseline characteristics of the study population by metabolic syndrome.

	Overall (*n* = 31,178)	No MetS (*n* = 21,660)	MetS (*n* = 8324)	SMD *
Age (years), mean (±SD)	50.1 (±11.1)	47.4 (±10.5)	57.4 (±9.1)	0.993
Age groups, *n* (%)				0.973
30–39 years	6893 (22.1)	6067 (28.0)	468 (5.6)	
40–49 years	7846 (25.2)	6488 (30.0)	1077 (12.9)	
50–59 years	8591 (27.6)	5659 (26.1)	2650 (31.8)	
60–69 years	7848 (25.2)	3446 (15.9)	4129 (49.6)	
BMI (kg/m^2^), mean (±SD)	23.6 (±3.6)	22.7 (±3.0)	26.1 (±3.7)	1.046
BMI groups, *n* (%)				0.967
<18.5	1304 (4.2)	1184 (5.5)	44 (0.5)	
18.5–24.9	20,447 (65.8)	16,322 (75.4)	3384 (40.7)	
25.0–29.9	7690 (24.7)	3744 (17.3)	3703 (44.6)	
30.0–34.9	1403 (4.5)	343 (1.6)	997 (12.0)	
≧35.0	246 (0.8)	43 (0.2)	183 (2.2)	
Education level, *n* (%)				0.757
≦Elementary	5484 (17.6)	2422 (11.2)	2851 (34.3)	
Middle school	3574 (11.5)	1972 (9.1)	1468 (17.7)	
High school	10,848 (34.8)	7853 (36.3)	2596 (31.2)	
≧College	11,240 (36.1)	9398 (43.4)	1395 (16.8)	
Income level, *n* (%)				0.162
Q1: Low	6046 (19.5)	3846 (17.8)	1929 (23.2)	
Q2: Low–middle	6229 (20.1)	4253 (19.7)	1736 (20.9)	
Q3: Middle	6240 (20.1)	4333 (20.1)	1690 (20.4)	
Q4: Middle–high	6280 (20.2)	4504 (20.9)	1566 (18.9)	
Q5: High	6246 (20.1)	4626 (21.5)	1376 (16.6)	
No Spouse, *n* (%)	4118 (14.0)	2326 (11.5)	1633 (20.2)	0.243
Smoking status, *n* (%)				0.075
Never smoker	27,693 (89.0)	19,209 (88.8)	7461 (89.9)	
Ex-smoker	1868 (6.0)	1368 (6.3)	396 (4.8)	
Current smoker	1560 (5.0)	1054 (4.9)	446 (5.4)	
Drinking habit, *n* (%)				0.290
Never drinking	3507 (11.3)	2745 (12.7)	653 (7.9)	
Normal drinking	23,350 (75.1)	16,516 (76.4)	5941 (71.6)	
High drinking	4235 (13.6)	2344 (10.8)	1706 (20.6)	
Walking days per week, *n* (%)				0.085
0 day	16,481 (52.9)	11,521 (53.2)	4347 (52.2)	
1–3 days	9770 (31.3)	6953 (32.1)	2442 (29.4)	
4–7 days	4917 (15.8)	3180 (14.7)	1531 (18.4)	
Exercise days per week, *n* (%)				0.165
0 day	25,437 (81.6)	17,303 (79.9)	7121 (85.6)	
1–3 days	3826 (12.3)	2976 (13.7)	743 (8.9)	
4–7 days	1912 (6.1)	1379 (6.4)	459 (5.5)	
Age of Menarche (years), mean (±SD)	14.1 (±2.0)	13.9 (±1.9)	14.7 (±2.1)	0.376
Pregnancies (times), mean (±SD)	3.2 (±1.9)	2.94 (±1.8)	3.74 (±1.9)	0.432
Age at First Delivery (years), mean (±SD)	26.1 (±4.1)	26.64 (±4.0)	24.73 (±3.9)	0.459
Number of Breastfed Children, mean (±SD)	1.3 (±1.8)	1.05 (±1.6)	1.90 (±2.2)	0.430
Post Menopause, *n* (%)	14,857 (47.7)	8159 (37.7)	6187 (74.3)	0.775
Used Oral Contraceptives, *n* (%)	4858 (15.6)	2928 (13.5)	1756 (21.1)	0.180

Abbreviations: BMI, body mass index; MetS, metabolic syndrome; SD, standard deviation; SMD, standardized mean difference. * SMD values of 0.2–0.5 are considered small effect sizes, values of 0.5–0.8 are considered medium effect sizes, and values > 0.8 are considered large effect sizes.

**Table 2 jcm-14-06319-t002:** Multivariate analysis of risk factors for metabolic syndrome by menopause status.

	Pre Menopause	Post Menopause
Variables	Adjusted OR (95% CI)^*^	Adjusted OR (95% CI) *
Age per 1 year	**1.11 (1.10–1.12)**	**1.09 (1.08–1.10)**
BMI per 1 kg/m^2^	**1.45 (1.42–1.48)**	**1.31 (1.29–1.33)**
Education level, College	1.00 (ref.)	1.00 (ref.)
High school	1.06 (0.90–1.25)	1.06 (0.91–1.23)
Middle school	1.12 (0.84–1.51)	1.15 (0.97–1.36)
Elementary	1.33 (0.96–1.85)	**1.19 (1.00–1.42)**
Income level, Q5: High	1.00 (ref.)	1.00 (ref.)
Q4: Middle–high	1.16 (0.93–1.46)	1.09 (0.95–1.26)
Q3: Middle	**1.26 (1.00–1.59)**	**1.17 (1.01–1.35)**
Q2: Low–middle	**1.31 (1.04–1.66)**	**1.20 (1.04–1.38)**
Q1: Low	**1.45 (1.14–1.84)**	**1.27 (1.10–1.48)**
Spouse, Yes	1.00 (ref.)	1.00 (ref.)
No	1.04 (0.81–1.33)	0.98 (0.87–1.10)
Smoking status, Never smoker	1.00 (ref.)	1.00 (ref.)
Ex-smoker	1.03 (0.77–1.38)	1.22 (0.95–1.55)
Current smoker	**1.42 (1.03–1.96)**	**1.60 (1.26–2.04)**
Drinking habit, Never drinking	1.00 (ref.)	1.00 (ref.)
Normal drinking	**1.28 (1.04–1.57)**	**1.29 (1.07–1.54)**
High drinking	**2.13 (1.58–2.88)**	**1.41 (1.16–1.73)**
Walking days per week, 0 days	1.00 (ref.)	1.00 (ref.)
1–3 days	1.15 (0.99–1.34)	1.05 (0.95–1.17)
4–7 days	0.98 (0.81–1.20)	1.08 (0.96–1.22)
Exercise days per week, 0 days	1.00 (ref.)	1.00 (ref.)
1–3 days	0.95 (0.76–1.19)	**0.77 (0.67–0.90)**
4–7 days	0.82 (0.61–1.10)	**0.62 (0.52–0.75)**
Age at Menarche per 1 year	1.00 (0.96–1.04)	0.99 (0.96–1.01)
Pregnancies per 1 time	0.98 (0.94–1.03)	1.00 (0.97–1.03)
Age at First Delivery per 1 year	0.99 (0.97–1.01)	0.99 (0.98–1.01)
Breast Feeding per 1 child	0.99 (0.93–1.04)	0.99 (0.96–1.02)
Oral Contraceptives, Not used	1.00 (ref.)	1.00 (ref.)
Used	**1.40 (1.13–1.72)**	**1.16 (1.03–1.29)**

***** Bolded numbers among adjusted ORs indicate statistical significance.

**Table 3 jcm-14-06319-t003:** Association between oral contraceptive duration and metabolic syndrome by menopausal status among 2010–2012 KNHANES participants.

Oral Contraceptive Duration	Pre Menopause		Post Menopause	
	MetS Cases/Sub Total Cases	Adjusted OR (95% CI) *	MetS Cases/Sub Total Cases	Adjusted OR (95% CI) *
Never used	350/3392	1.00	995/2590	1.00
<1 year	16/180	0.97 (0.44–2.16)	119/263	1.13 (0.79–1.63)
1–3 years	19/96	1.48 (0.72–2.84)	151/321	1.12 (0.80–1.58)
3–5 years	2/19	1.50 (0.06–40.62)	65/106	1.26 (0.73–2.16)
≧5 years	3/26	0.44 (0.13–1.49)	48/76	**2.13 (1.27–3.57)**

* Adjusted ORs were estimated from models of adjusted covariates (age, BMI, number of pregnancies, age at first delivery, breastfeeding, age at menarche, education level, income level, spousal status, smoking status, drinking habits, walking, and exercise). Bolded numbers among adjusted ORs indicate statistical significance.

## Data Availability

The data used in this study are publicly available and can be freely downloaded from the KNHANES website (https://knhanes.kdca.go.kr/ (accessed on 1 August 2025)).

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
