# Peer review of "Oral Contraceptive Use and Reproductive History in Relation to Metabolic Syndrome Among Women from KNHANES 2010–2023"

_jcm, 2025, doi:10.3390/jcm14176319_

Round 1
Reviewer 1 Report (Previous Reviewer 2)
Comments and Suggestions for Authors
The manuscript addresses a relevant and underexplored question in the Asian literature: the association between oral contraceptive (OC) use and metabolic syndrome (MetS), stratified by menopausal status, using a large, nationally representative sample from KNHANES. The use of complex survey weights, the large sample size, and the stratification by individual MetS components are notable strengths. The spline modelling further enriches the analysis.
To further improve clarity and precision before acceptance, I recommend addressing the following minor points:
-
Clarify in Methods the operational definition of menopausal status and how surgical menopause or hormone replacement therapy cases were classified.
-
Review and correct potential typos or duplicated values in Table 2 (e.g., identical ORs for two strength‐exercise categories).
-
Briefly describe in the text or a figure legend the procedure for combining 2010–2023 survey weights, citing the official guidance used.
-
Maintain consistent terminology throughout the manuscript (e.g., always “KNHANES”).
-
In the Conclusions, slightly adjust the wording to reflect association rather than causality.
With these small refinements, the manuscript will be ready for publication.
Author Response
Reviewer 1)
The manuscript addresses a relevant and underexplored question in the Asian literature: the association between oral contraceptive (OC) use and metabolic syndrome (MetS), stratified by menopausal status, using a large, nationally representative sample from KNHANES. The use of complex survey weights, the large sample size, and the stratification by individual MetS components are notable strengths. The spline modelling further enriches the analysis.
To further improve clarity and precision before acceptance, I recommend addressing the following minor points:
- Clarify in Methods the operational definition of menopausal status and how surgical menopause or hormone replacement therapy cases were classified.
Response) We thank the reviewer for this valuable comment. We have revised the Methods section to clarify the operational definition of menopausal status and the classification of surgical menopause and hormone replacement therapy (HRT) cases. Specifically, menopausal status was determined using the survey question “Are you currently having menstruation?” with the response options “natural menopause” or “artificial menopause.” Women who selected either option and additionally reported their age at menopause were classified as postmenopausal, while those reporting ongoing menstruation were considered premenopausal. The questionnaire did not collect further details regarding the reasons for artificial menopause (e.g., bilateral oophorectomy, hysterectomy, or other causes). Information on HRT use was not available in the dataset and thus was not considered in the classification. This clarification has been added to the Methods section (lines 89–96).
- Review and correct potential typos or duplicated values in Table 2 (e.g., identical ORs for two strength‐exercise categories).
Response) We appreciate the reviewer’s careful attention to detail. We rechecked the calculations in Table 2 and found duplicated values in the results for the walking and exercise categories in the postmenopausal group. Table 2 has now been corrected with the accurate odds ratios and 95% confidence intervals.
- Briefly describe in the text or a figure legend the procedure for combining 2010–2023 survey weights, citing the official guidance used.
Response) We thank the reviewer for this helpful suggestion. We have revised the manuscript to briefly describe the procedure used for combining survey weights across the 2010–2023 KNHANES waves. As recommended in the official KNHANES guidelines, the integrated weights were calculated by dividing each individual survey weight by the number of combined survey years. This procedure has now been added to the Methods section (lines 137–141) and cited.
- Maintain consistent terminology throughout the manuscript (e.g., always “KNHANES”).
Response) We appreciate this helpful comment. We have revised the manuscript to ensure consistent terminology, using the full name "Korea National Health and Nutrition Examination Survey (KNHANES)" only at first mention and the abbreviation "KNHANES" thereafter. We also revised the title, replacing “Korea NHANES” with the consistent abbreviation “KNHANES.
- In the Conclusions, slightly adjust the wording to reflect association rather than causality.
Response) We appreciate the reviewer’s important comment. We carefully reviewed the Conclusions section and confirmed that the wording already emphasizes associations rather than causality. For further clarity, we made minor adjustments to reinforce this point. Specifically, we ensured that terms such as “associated with” and “observational findings” are consistently used, and avoided any phrasing that could imply a causal relationship.
With these small refinements, the manuscript will be ready for publication.
Reviewer 2 Report (New Reviewer)
Comments and Suggestions for Authors
This study investigated the association between reproductive factors and the presence of MetS in the cross-sectional manner and found that usage of oral contraceptive was associated with the presence of MetS. Although the cross-sectional design was a weak of this study, this article was well written and easy to follow. This reviewer has a one question.
1. As previously reported, it may be worth considering the impact of menopause on the presence of MetS.
Author Response
Reviewer 2)
This study investigated the association between reproductive factors and the presence of MetS in the cross-sectional manner and found that usage of oral contraceptive was associated with the presence of MetS. Although the cross-sectional design was a weak of this study, this article was well written and easy to follow. This reviewer has a one question.
- As previously reported, it may be worth considering the impact of menopause on the presence of MetS.
Response) We thank the reviewer for this insightful comment. We fully agree that menopausal status is an important factor in relation to MetS. In our study, a similar pattern was observed: as shown in Table 1, although not adjusted some covariates such as age, BMI, and etc., the prevalence of MetS was 74.3% among postmenopausal women compared with 37.7% among premenopausal women. Therefore, to account for the impact of menopause, we stratified our analyses by menopausal status when examining the association between OC use and MetS. This approach allowed us to account for the impact of menopause on MetS, which was one of the key objectives of our study.
This manuscript is a resubmission of an earlier submission. The following is a list of the peer review reports and author responses from that submission.
Round 1
Reviewer 1 Report
Comments and Suggestions for Authors
Based on a careful read of the manuscript titled "Oral Contraceptive Use and Reproductive History in Relation to Metabolic Syndrome Among Women from Korea NHANES 2010–2023", I would like to offer a few minor but constructive suggestions that could help enhance the clarity, depth, and presentation of the study.
First, in the Methods section, it would be helpful to clarify how “oral contraceptive use” was defined. Was it based on any prior use (“ever used”), or was there a minimum required duration or recency of use? This detail is important for interpretation, especially since the metabolic impact of OCs can vary by formulation, dose, and duration. Relatedly, while the authors acknowledge the lack of OC formulation details as a limitation, even a rough breakdown by decade of use or typical formulations available in Korea during the study period would provide valuable context. That could also help support the discussion of possible estrogen- or progestin-related mechanisms.
Since this is a cross-sectional study, I would suggest briefly addressing the potential for reverse causality—for example, women with existing metabolic concerns might be less likely to initiate or may discontinue OC use. While this may not fully explain the observed association, acknowledging it could help preempt future critiques.
The authors note that reproductive factors such as age at menarche or parity were not independently associated with MetS after adjustment. It might be useful to explore whether this lack of association could be influenced by collinearity with variables like age or BMI, or whether these reproductive factors are simply less predictive in contemporary populations with generally improved maternal and perinatal care.
Additionally, the finding that strength exercise was inversely associated with MetS in postmenopausal women, but not in premenopausal women, is quite interesting. A brief hypothesis—such as the possible greater metabolic benefits of resistance training in sarcopenia-prone older women—could strengthen the implications of this finding.
Figures 2 and 3 are overall clear and informative. That said, adding clearer axis labels (e.g., explicitly indicating that 1.0 refers to the reference point for OR) and defining abbreviations such as “MetS” and “OC” directly in the figure legends would improve their standalone readability.
A few minor editorial points:
-
Line 190: “Middel-high” should be corrected to “Middle-high.”
-
Line 338: If the 2023 research grant refers to support for a retrospective study covering 2010–2023, you might briefly clarify its scope.
-
Throughout the manuscript, please check for consistency in the use of “OC” vs. “oral contraceptives” (especially between figures and main text).
These suggestions are mostly stylistic or clarifying in nature, and I commend the authors for a well-structured and data-rich manuscript on an important topic in women’s metabolic health.
Author Response
Author’s Response
[Reviewer 1]
Based on a careful read of the manuscript titled "Oral Contraceptive Use and Reproductive History in Relation to Metabolic Syndrome Among Women from Korea NHANES 2010–2023", I would like to offer a few minor but constructive suggestions that could help enhance the clarity, depth, and presentation of the study.
First, in the Methods section, it would be helpful to clarify how “oral contraceptive use” was defined. Was it based on any prior use (“ever used”), or was there a minimum required duration or recency of use? This detail is important for interpretation, especially since the metabolic impact of OCs can vary by formulation, dose, and duration. Relatedly, while the authors acknowledge the lack of OC formulation details as a limitation, even a rough breakdown by decade of use or typical formulations available in Korea during the study period would provide valuable context. That could also help support the discussion of possible estrogen- or progestin-related mechanisms.
- We appreciate the reviewer’s insightful comment regarding the definition of oral contraceptive (OC) use. In the KNHANES dataset, OC use was assessed through the following standardized question: “Have you ever taken oral contraceptive pills for more than one month?” Therefore, OC use in our study was defined as ever having used oral contraceptives for at least one month. Unfortunately, detailed information on the total duration, timing, or specific formulations of OC use was only collected during 2010–2012 and was not available for the 2013–2023 study period. As such, we were unable to incorporate dose, formulation, or duration into our analysis. We have revised the Methods section to clarify this definition and added this point as a limitation in the Discussion.
[Revised Methods Section: Lines 111-113]
“OC use was defined based on a standardized question: ‘Have you ever taken oral contraceptive pills for more than one month?’ Those who answered ‘yes’ were classified as OC users.”
[Revised Discussion Section: Lines 328-332]
“Although formulation-specific data were not available in KNHANES, most oral contraceptives marketed in Korea during the 2010s included low-dose combined estrogen-progestin pills, with ethinyl estradiol typically at 20–30 µg. Therefore, our findings may reflect the metabolic impact of second- and third-generation combined OCs commonly used in this region and era.”
Since this is a cross-sectional study, I would suggest briefly addressing the potential for reverse causality—for example, women with existing metabolic concerns might be less likely to initiate or may discontinue OC use. While this may not fully explain the observed association, acknowledging it could help preempt future critiques.
- We thank you for this thoughtful and constructive suggestion. We agree that the possibility of reverse causality is an important consideration in interpreting the findings from a cross-sectional study design. In response, we have revised the Discussion section to explicitly acknowledge this limitation. Specifically, we added the following sentences to the first limitation paragraph in the Discussion:
[Revised Discussion Section: Lines 317-319]
“Additionally, we cannot rule out the possibility of reverse causality; for example, women with existing metabolic risk or conditions may have been less likely to initiate or may have discontinued OC use. Therefore, our findings should be interpreted as associations rather than causal links.”
The authors note that reproductive factors such as age at menarche or parity were not independently associated with MetS after adjustment. It might be useful to explore whether this lack of association could be influenced by collinearity with variables like age or BMI, or whether these reproductive factors are simply less predictive in contemporary populations with generally improved maternal and perinatal care.
- As suggested, we agree that the lack of significant associations between certain reproductive factors and MetS in the adjusted models may be partly attributable to collinearity with other variables such as age or BMI, which are both strongly associated with metabolic risk. To assess this, we checked the variance inflation factor (VIF) for all covariates, and found no evidence of problematic multicollinearity (VIFs < 2.0), suggesting that collinearity may not fully explain the null findings. We also acknowledge the possibility that improvements in maternal and perinatal care, as well as overall population health trends, may have attenuated the impact of reproductive exposures on metabolic outcomes in contemporary cohorts. We have added a brief discussion of these points to the revised manuscript.
[Revised Discussion Section: Lines 272-277]
“The lack of associations may also result from collinearity with age or BMI, both of which are strong predictors of MetS [22]. However, in this study, variance inflation factors were <2.0, suggesting no serious multicollinearity. Another possibility is that improvements in maternal and women's health care over recent decades have diminished the predictive value of these reproductive markers [23].”
Additionally, the finding that strength exercise was inversely associated with MetS in postmenopausal women, but not in premenopausal women, is quite interesting. A brief hypothesis—such as the possible greater metabolic benefits of resistance training in sarcopenia-prone older women—could strengthen the implications of this finding.
- We thank the reviewer for this insightful suggestion. We agree that the inverse association between strength exercise and MetS observed only in postmenopausal women warrants further discussion. In response, we have added a brief hypothesis in the Discussion section. Specifically, we noted that postmenopausal women may derive greater metabolic benefit from resistance training due to age-related changes in body composition, including increased risk of sarcopenia and visceral fat accumulation. These changes may render postmenopausal women more responsive to strength-based interventions for metabolic health. This addition helps contextualize the differential association observed by menopausal status.
[Revised Discussion Section: Lines 302-306]
“Finally, strength exercise was inversely associated with MetS only among postmenopausal women. This may reflect greater metabolic benefits of resistance training in older women, who are more susceptible to sarcopenia and visceral fat accumulation after menopause [33]. These findings suggest that incorporating strength-based interventions could be particularly effective in mitigating metabolic risk in this population.”
Figures 2 and 3 are overall clear and informative. That said, adding clearer axis labels (e.g., explicitly indicating that 1.0 refers to the reference point for OR) and defining abbreviations such as “MetS” and “OC” directly in the figure legends would improve their standalone readability.
- We have updated the figure 2 and 3 legends to define all abbreviations, including “MetS” (metabolic syndrome) and “HDL” (high-density lipoprotein), to enhance the standalone readability and interpretability of the figures.
A few minor editorial points:
- Line 190: “Middel-high” should be corrected to “Middle-high.”
- The typo ‘Middel-high’ in Table 1 and 2 has been corrected to ‘Middle-high’.
- Line 338: If the 2023 research grant refers to support for a retrospective study covering 2010–2023, you might briefly clarify its scope.
- The 2023 research grant from Gyeongsang National University was not specifically tied to the time frame of the data used (2010–2023), but rather supported the overall process of any research. Therefore, we believe that revision is not necessary as the current funding statement does not create confusion about the scope of support.
- Throughout the manuscript, please check for consistency in the use of “OC” vs. “oral contraceptives” (especially between figures and main text).
- We have reviewed the manuscript for consistency in terminology. In accordance with standard conventions, we used the full term “oral contraceptives” in subheadings for clarity, while using the abbreviation “OC” in the main text and figure legends after its first mention.
These suggestions are mostly stylistic or clarifying in nature, and I commend the authors for a well-structured and data-rich manuscript on an important topic in women’s metabolic health.
- We appreciate your review of the positive evaluation and constructive suggestions. Your comments helped us improve the clarity and presentation of the manuscript.
[Reviewer 2]
The manuscript analyses fourteen consecutive waves of KNHANES (>31,000 women aged 30-69 years) and concludes that oral-contraceptive (OC) use—rather than other reproductive factors is independently associated with metabolic-syndrome (MetS) in Korean women, mainly through blood-pressure and triglyceride pathways. The sample is large, nationally representative and, by stratifying by menopausal status, the study answers a question of genuine clinical relevance. That said, several methodological details, interpretative gaps and bibliographic weaknesses must be addressed.
Major issues that need attention
- Literature coverage.
Twenty-seven references are too few for a paper that ranges across MetS epidemiology, reproductive history, hormonal contraception, menopausal transition and Korean health trends. Sixteen of those citations sit almost entirely in the Introduction, leaving the Discussion to lean on just eleven sources. One paper (Bai et al.) appears twice—dated 2022 and 2023; please keep one entry (print year 2023). I would also expect to see recent cohort or meta-analysis work on low-dose OCs and MetS (e.g. Circulation 2023, BMJ 2022), contemporary Asian trend data, and guideline documents such as WHO MEC 2022 or the latest FIGO/ACOG statements.
- We appreciate your suggestions regarding additional references. However, the primary aim of our study was to examine the association between reproductive factors, including oral contraceptive (OC) use, and metabolic syndrome (MetS), rather than cardiovascular or thrombotic outcomes. Mentioned references (BMJ 2022) focus on the risk of arterial or venous thrombotic events related to OC use, which reflect a distinct pathophysiological pathway compared to metabolic dysfunction. We carefully reviewed the suggested articles and found that their direct applicability to the MetS context is limited. Instead of incorporating those references, we strengthened our discussion by focusing on prior literature directly addressing metabolic outcomes of hormonal contraception. Where appropriate, we acknowledged broader cardiometabolic implications - for example, citing the 2023 UK Biobank cohort study and the 2024 U.S. Medical Eligibility Criteria, which recommend individualized contraceptive risk assessment for women with metabolic risk factors.
- We have also corrected the duplication of the Bai et al. reference and retained only the 2023 publication year.
[Revised Discussion Section: Lines 291-301]
“While our findings suggest a modest but consistent increase in MetS risk associated with OC use, the literature offers mixed evidence. For example, a 2023 cohort study using UK Biobank found no increased CVD or mortality risk from OC use and even suggested protective effects among long-term users [31]. However, that study focused on different outcomes and included mainly White British women, limiting relevance to our population.
In our analysis, OC use was not associated with stricter definitions of MetS or with glucose and waist circumference. This suggests that OC use may selectively affect certain metabolic domains, rather than causing global dysfunction. These findings highlight the importance of incorporating reproductive and hormonal history into metabolic risk assessment. International guidelines, such as the 2024 U.S. Medical Eligibility Criteria, recommend caution with OC use in women with cardiometabolic risk factors [32].”
KNHANES records only a binary “ever used OC”. If you have any information on duration, generation or age at first/last use, please say so; if not, flag the limitation explicitly. A sensitivity analysis limited to younger women, or to those who used OCs within the past decade, would help gauge misclassification.
- KNHANES collects data only on whether participants have ever used oral contraceptives (OC) for more than one month, without information on the duration, type, or timing of use. We have revised the Methods section to clarify this point. In addition, we have acknowledged this limitation more explicitly in the Discussion, noting that the inability to assess detailed OC exposure may limit the interpretation of our findings.
[Revised Methods Section: Lines 111-113]
“OC use was defined based on a standardized question: ‘Have you ever taken oral contraceptive pills for more than one month?’ Those who answered ‘yes’ were classified as OC users.”
[Revised Discussion Section: Lines 328-332]
“Although formulation-specific data were not available in KNHANES, most oral contraceptives marketed in Korea during the 2010s included low-dose combined estrogen-progestin pills, with ethinyl estradiol typically at 20–30 µg. Therefore, our findings may reflect the metabolic impact of second- and third-generation combined OCs commonly used in this region and era.”
- Missing data.
The flow-chart shows exclusions for reproductive variables but not for missing BMI, lipids, lifestyle factors, etc. Readers need to know what proportion was lost for each covariate, whether list-wise deletion or multiple imputation was used, and how survey weights were adjusted after exclusions.
- We appreciate this important comment. We assessed the missingness of all covariates used in the multivariable models, including BMI, lipid profiles, age, and lifestyle factors. The overall proportion of missing data was low; most variables had <3% missing values, with the highest being 5.78% for marital status. Given this low level of missingness, we did not perform multiple imputation and instead conducted complete case analyses for covariates. These missing data proportions are now presented in a supplementary table (Supplementary Table 1), and corresponding details have been added to the Methods section.
[Revised Methods Section: Lines 114-116]
“Covariates with missing values were not imputed and were analyzed as observed. The proportion of missing data for each covariate is provided in the Supplementary Table 1.”
- Spline details.
Please state how many knots you used, where they were placed, provide a goodness-of-fit statistic and—ideally—post the R code or knot coordinates in the Supplement. Otherwise, the non-linear curves cannot be reproduced.
- We appreciate this important comment. We used four knots in the restricted cubic spline (RCS) model, which were placed at the default percentiles—specifically the 5th, 35th, 65th, and 95th percentiles—of the exposure variable distribution. These knot locations have now been explicitly described in the revised Methods section of the manuscript. To present the model’s goodness-of-fit, we added both the pseudo R² (Nagelkerke’s R²) and the C-statistic values directly to the figure. Descriptions of these statistical metrics have also been incorporated into the Statistical Analysis section.
- Additionally, the full R code used for the RCS modeling, including knot specification and plotting, has been provided in the Supplementary Material to ensure transparency and reproducibility of our analysis. The figure legend has been revised accordingly to reflect the number and placement of knots as well as the reported model fit indices.
[Revised Methods Section: Lines 148-153]
“We used four knots placed at the 5th, 35th, 65th, and 95th percentiles of each exposure distribution, based on default quantile placement. Goodness-of-fit for each RCS model was assessed using Nagelkerke’s pseudo R² and the C-statistic, and these values were presented on the corresponding spline plots.”
[Revised Methods Section: Lines 165-166]
“The full R code used for all analysis in this study is provided in the Supplementary Material for reproducibility.”
[Also, please find the Revised Figure 2]
- Causal wording.
A cross-sectional design cannot show that OC use leads to or results in hypertension. Phrases like that should be replaced with “is associated with”, unless you have longitudinal evidence to justify stronger language.
- We have revised the manuscript to consistently describe the relationship between OC use and metabolic syndrome or hypertension using non-causal language such as “is associated with” or “may be linked to.” Additionally, we have clarified in the limitations section that the cross-sectional nature of the study precludes establishing causal relationships.
[Revised Discussion Section: Lines 317-319]
“Additionally, we cannot rule out the possibility of reverse causality; for example, women with existing metabolic risk or conditions may have been less likely to initiate or may have discontinued OC use. Therefore, our findings should be interpreted as associations rather than causal links.”
Minor issues that need attention
Introduction: The mechanistic paragraphs rest on one or two studies each, some of them from the 1990s. Please weave in more recent lipidomic and haemodynamic work that uses modern formulations.
- We appreciate your comment regarding the mechanistic discussion. While the references cited in the introduction were published after 2000, we agree that the biological mechanisms linking OC use to cardiometabolic risk were not sufficiently detailed. Therefore, we have revised the Introduction and Discussion sections to include more recent studies, particularly lipidomic and hemodynamic research using modern oral contraceptive formulations. These additions help to clarify the plausible pathways by which OC use may influence MetS risk.
[Revised Introduction Section: Lines 61-63]
“Recent lipidomic and hemodynamic studies show that modern OC formulations may affect vascular resistance, insulin sensitivity, and lipid metabolism, potentially increasing cardiometabolic risk [17,18].”
Results: Tables are informative but crowded. Bold significant adjusted ORs and consider moving crude ORs to the Supplement to streamline the narrative.
- In response to your comment, we moved the crude ORs from Table 2 to Supplementary Table 2 to streamline the presentation. In addition, we bolded statistically significant adjusted ORs in Table 2 to enhance clarity and interpretability.
Discussion: It would help to (1) set your null result for age at menarche against cohorts that report a reversed-J curve, and discuss possible ethnic or adiposity modifiers; and (2) spell out what your findings mean for everyday care—should Korean clinicians screen earlier for MetS in women with an OC history?
- We appreciate the reviewer’s insightful suggestions. First, we have added a comparison of our null findings regarding age at menarche with prior studies that reported a reversed J-shaped association. We also discussed possible explanations for these differences, including ethnic variability in reproductive timing, genetic predispositions, and the modifying effect of adiposity and body composition. These points have been incorporated into the revised Discussion section.
- Second, to address the clinical implications of our findings, we have added a paragraph discussing how the observed association between OC use and MetS, particularly in postmenopausal women, may inform risk stratification in clinical settings. Specifically, we suggest that clinicians in Korea may consider earlier screening or more vigilant monitoring for MetS in women with a history of prolonged OC use. This recommendation has also been added to the Discussion.
[Revised Discussion Section: Lines 271-277]
“Additionally, population-specific factors such as genetics, diet, or cultural norms may affect these associations [21]. The lack of associations may also result from collinearity with age or BMI, both of which are strong predictors of MetS [22]. However, in this study, variance inflation factors were <2.0, suggesting no serious multicollinearity. Another possibility is that improvements in maternal and women's health care over recent decades have diminished the predictive value of these reproductive markers [23].”
[Revised Discussion Section: Lines 298-301]
“These findings highlight the importance of incorporating reproductive and hormonal history into metabolic risk assessment. International guidelines, such as the 2024 U.S. Medical Eligibility Criteria, recommend caution with OC use in women with cardiometabolic risk factors [32].”
English style: Many sentences run over forty words or contain double qualifiers. A light professional edit to shorten and standardise tense would make the paper easier to follow.
- In response to the comment, we thoroughly reviewed the manuscript and edited the sentences to enhance clarity by shortening overly long constructions and eliminating double qualifiers where appropriate.
Figures: A footnote reminding readers that the estimates are survey-weighted—and noting knot positions in Figure 2—would be welcome. The labels in Figure 3 are almost unreadable on print-out; please enlarge them.
- Figures 2 and 3 have been revised per your comments. Figure 2 has been updated and the title of Figure 3 has been bolded. Please let us know if any further revisions are required.
Questions.
- Did you test interaction terms between OC use and BMI or smoking?
- Yes, we tested interaction terms between OC use and both BMI and smoking status using survey-weighted logistic regression models. However, none of the interaction terms reached statistical significance (e.g., p=0.28 for OC×BMI, p=0.73 for OC×smoking), suggesting no meaningful effect modification. As these interactions were not statistically significant and did not alter the primary findings, we did not include them in the main manuscript to maintain focus and clarity. The R code was as follows:
> summary(model_interaction)
Call:
svyglm(formula = metabolic_syndrome ~ oc * bmi + oc * smoking +
age + edu + incm5 + spouse + drinking + walking + exercise +
age_ms + pregnancy + age_delivery + b_feeding + menopause,
design = survey_design, family = quasibinomial())
Survey design:
svydesign(id = ~psu, strata = ~kstrata, weights = ~weight, data = all_model, nest = TRUE)
- Given that OC formulations changed over 2009-2023, did you check for calendar-period interactions?
- We appreciate this insightful comment. However, we did not perform calendar-period interaction analyses for two main reasons. First, the KNHANES dataset does not provide detailed information regarding the formulation, dosage, or generation of oral contraceptives used by participants, limiting our ability to assess changes in OC characteristics over time. Second, evaluating temporal shifts in OC composition and their differential metabolic effects was beyond the primary scope of our study, which aimed to examine the overall association between OC use and metabolic syndrome (MetS). We agree that future studies with formulation-specific exposure data and longitudinal follow-up would be better suited to address this important question.
- Why did you keep the ATP III “≥ 3 criteria” definition when several Asian guidelines suggest ethnicity-specific waist-circumference cut-points?Reconsider after major revisions.
- Thank you for this important observation. We used the ATP III criteria, including the ≥88 cm waist circumference threshold for women, to ensure consistency with the majority of prior Korean and international studies using KNHANES data, thereby enhancing comparability of our findings. While we acknowledge that alternative guidelines (e.g., IDF, WHO WPRO) propose lower ethnicity-specific cutoffs (e.g., ≥80 cm for Asian women), the ATP III definition remains widely used in Korean population-based research and national reports, including by the Korea Disease Control and Prevention Agency.
Reconsider after major revisions. The dataset is rich and the menopausal-stratified analysis original but it needs (1) a broader, up-to-date bibliography, (2) fuller methodological transparency and (3) a more concise narrative. I would be happy to look at a thoroughly revised version.
Comments on the Quality of English Language
The manuscript is generally intelligible; however, the prose is heavy. Many sentences exceed 35–40 words, contain duplicate modifiers (e.g., "significantly higher increase"), or string multiple clauses together without clear punctuation.
- In response to the comment, we thoroughly reviewed the manuscript and edited the sentences to enhance clarity by shortening overly long constructions and eliminating double qualifiers where appropriate.

Reviewer 2 Report
Comments and Suggestions for Authors
The manuscript analyses fourteen consecutive waves of KNHANES (>31,000 women aged 30-69 years) and concludes that oral-contraceptive (OC) use—rather than other reproductive factors is independently associated with metabolic-syndrome (MetS) in Korean women, mainly through blood-pressure and triglyceride pathways. The sample is large, nationally representative and, by stratifying by menopausal status, the study answers a question of genuine clinical relevance. That said, several methodological details, interpretative gaps and bibliographic weaknesses must be addressed.
Major issues that need attention
- Literature coverage.
Twenty-seven references are too few for a paper that ranges across MetS epidemiology, reproductive history, hormonal contraception, menopausal transition and Korean health trends. Sixteen of those citations sit almost entirely in the Introduction, leaving the Discussion to lean on just eleven sources. One paper (Bai et al.) appears twice—dated 2022 and 2023; please keep one entry (print year 2023). I would also expect to see recent cohort or meta-analysis work on low-dose OCs and MetS (e.g. Circulation 2023, BMJ 2022), contemporary Asian trend data, and guideline documents such as WHO MEC 2022 or the latest FIGO/ACOG statements.
- Definition of OC exposure.
KNHANES records only a binary “ever used OC”. If you have any information on duration, generation or age at first/last use, please say so; if not, flag the limitation explicitly. A sensitivity analysis limited to younger women, or to those who used OCs within the past decade, would help gauge misclassification.
- Missing data.
The flow-chart shows exclusions for reproductive variables but not for missing BMI, lipids, lifestyle factors, etc. Readers need to know what proportion was lost for each covariate, whether list-wise deletion or multiple imputation was used, and how survey weights were adjusted after exclusions.
- Spline details.
Please state how many knots you used, where they were placed, provide a goodness-of-fit statistic and—ideally—post the R code or knot coordinates in the Supplement. Otherwise, the non-linear curves cannot be reproduced.
- Causal wording.
A cross-sectional design cannot show that OC use leads to or results in hypertension. Phrases like that should be replaced with “is associated with”, unless you have longitudinal evidence to justify stronger language.
Minor issues that need attention
Introduction: The mechanistic paragraphs rest on one or two studies each, some of them from the 1990s. Please weave in more recent lipidomic and haemodynamic work that uses modern formulations.
Results: Tables are informative but crowded. Bold significant adjusted ORs and consider moving crude ORs to the Supplement to streamline the narrative.
Discussion: It would help to (1) set your null result for age at menarche against cohorts that report a reversed-J curve, and discuss possible ethnic or adiposity modifiers; and (2) spell out what your findings mean for everyday care—should Korean clinicians screen earlier for MetS in women with an OC history?
English style: Many sentences run over forty words or contain double qualifiers. A light professional edit to shorten and standardise tense would make the paper easier to follow.
Figures: A footnote reminding readers that the estimates are survey-weighted—and noting knot positions in Figure 2—would be welcome. The labels in Figure 3 are almost unreadable on print-out; please enlarge them.
Questions.
- Did you test interaction terms between OC use and BMI or smoking?
- Given that OC formulations changed over 2009-2023, did you check for calendar-period interactions?
3.Why did you keep the ATP III “≥ 3 criteria” definition when several Asian guidelines suggest ethnicity-specific waist-circumference cut-points?Reconsider after major revisions.
Reconsider after major revisions. The dataset is rich and the menopausal-stratified analysis original but it needs (1) a broader, up-to-date bibliography, (2) fuller methodological transparency and (3) a more concise narrative. I would be happy to look at a thoroughly revised version.
Comments on the Quality of English LanguageThe manuscript is generally intelligible; however, the prose is heavy. Many sentences exceed 35–40 words, contain duplicate modifiers (e.g., "significantly higher increase"), or string multiple clauses together without clear punctuation.
Author Response

(The authors gave the same response as above.)

Round 2
Reviewer 2 Report
Comments and Suggestions for Authors
Thank you for your detailed point-by-point reply and the improvements made to the manuscript. You clarified the definition of oral contraceptive (OC) exposure, specified spline knot placement, added interaction terms, and expanded the literature somewhat. However, despite these changes, several fundamental issues remain unresolved, and the revised version does not meet the scientific standards required for publication.
The key reasons for this recommendation are:
- Exposure misclassification.
OC use remains classified as a binary “ever used for ≥1 month,” without consideration for duration, formulation generation, or recency. This leads to substantial exposure misclassification and limits interpretability. The requested sensitivity analyses (e.g., by duration using 2010–2012 data, or limited to younger women likely to use low-dose OCs) were not performed. - Outcome definition—waist circumference cut-points.
Your main analyses use the ATP III definition (waist ≥88 cm). Yet, for Asian populations, WHO and IDF guidelines recommend ≥80 cm. A robustness check using this alternative cut-point is essential and still missing. - Handling of missing data.
Although you now report the percentage of excluded cases, you continue to use listwise deletion without multiple imputation or reweighting. This may introduce bias and reduce generalizability. - Overstated interpretation.
Despite some toned-down wording, the Abstract and Conclusions still imply causality and public health actionability not supported by the cross-sectional design. Statements such as “should be considered in screening” remain inappropriate. - Literature remains insufficient.
Although you expanded your references, critical modern cohort studies and key clinical guidelines (e.g., US MEC 2024, IDF MetS criteria for Asian populations) are either not cited or not meaningfully integrated into the Discussion.
Your study addresses a relevant question and is based on a rich dataset. However, the methodological limitations, particularly regarding exposure definition, sensitivity analysis, and incomplete robustness checks, prevent firm conclusions from being drawn. As currently presented, the manuscript does not meet the standards for publication in a high-impact clinical journal. A substantially restructured and reanalyzed manuscript would be required for further consideration.
Comments on the Quality of English LanguageOverall the English is clear and intelligible, but a light copy-edit would still help. A few sentences run over forty words, some duplicative qualifiers remain (“significantly higher increase”), and minor punctuation inconsistencies appear in the new reference list. Tightening these items and standardising tense will further improve flow and readability.
Author Response
The key reasons for this recommendation are:
- Exposure misclassification.
OC use remains classified as a binary “ever used for ≥1 month,” without consideration for duration, formulation generation, or recency. This leads to substantial exposure misclassification and limits interpretability. The requested sensitivity analyses (e.g., by duration using 2010–2012 data, or limited to younger women likely to use low-dose OCs) were not performed.
à We thank the reviewer for this important feedback regarding OC exposure classification. We have addressed these concerns through the following revisions:
As requested, we have now performed sensitivity analyses examining the association between OC duration and Metabolic Syndrome using 2010–2012 data. Due to small sample sizes in some duration categories that precluded meaningful linear trend analysis, we categorized OC use as: never used, <1 year, 1–3 years, 3–5 years, and >5 years. The methodology for this analysis has been added to the Methods section, and the results are presented in Table 3 and discussed in the Discussion section.
- Outcome definition—waist circumference cut-points.
Your main analyses use the ATP III definition (waist ≥88 cm). Yet, for Asian populations, WHO and IDF guidelines recommend ≥80 cm. A robustness check using this alternative cut-point is essential and still missing.
à Although international guidelines suggest ≥80 cm as obesity, actual diagnostic criteria vary considerably across Asian countries. The Korean Society for the Study of Obesity recommends ≥85 cm, the Japan Society for the Study of Obesity uses ≥90 cm, and the Chinese Diabetes Society applies ≥85 cm for waist circumference thresholds.
Ref) Korean Society for Obesity. Obesity Fact Sheet 2023. https://www.kosso.or.kr/popup/obesity_fact_sheet.html
Ref) Japanese Society for the Study of Obesity Guidelines for the Management of Obesity Disease 2022 https://www.jstage.jst.go.jp/article/endocrj/71/3/71_EJ23-0593/_html/-char/en?utm_source=chatgpt.com
Ref) Chinese Diabetes Society. Guideline for the prevention and treatment of diabetes mellitus in China (2024 edition). https://diab.cma.org.cn/cn/zhinangongshi.aspx
These variations reflect population-specific considerations and validation studies within each country. Given this heterogeneity in country-specific guidelines and to maintain consistency with the majority of published MetS research, we elected to use the widely-adopted ATP III criterion of ≥88 cm for our primary analysis. This approach facilitates comparison with the extensive body of existing literature on MetS and hormonal contraceptive use.
Nevertheless, recognizing the importance of the IDF recommendation, we have conducted the requested sensitivity analysis using the ≥80 cm cutoff. As presented in Supplemental Table 3, the results using this alternative threshold are consistent with our primary findings, demonstrating the robustness of our conclusions across different diagnostic criteria.
- Handling of missing data.
Although you now report the percentage of excluded cases, you continue to use listwise deletion without multiple imputation or reweighting. This may introduce bias and reduce generalizability.
à Our study has a very low missing data rate of <6%, which falls well below the commonly accepted threshold of 10% where missing data handling techniques become critical. Given this minimal level of missingness, the risk of bias introduced by listwise deletion is likely negligible. Moreover, the use of more complex methods, such as multiple imputation or weighting, may not offer substantial advantages and could potentially introduce bias when applied inappropriately to datasets with low levels of missing data.
- Overstated interpretation.
Despite some toned-down wording, the Abstract and Conclusions still imply causality and public health actionability not supported by the cross-sectional design. Statements such as “should be considered in screening” remain inappropriate.
à We have comprehensively reviewed the entire manuscript, with particular attention to the Abstract and Discussion sections, to address this concern. Throughout the revised manuscript, we have emphasized the cross-sectional nature of our study and carefully modified our language to focus on associations between OC use and MetS rather than implying causality.
- Literature remains insufficient.
Although you expanded your references, critical modern cohort studies and key clinical guidelines (e.g., US MEC 2024, IDF MetS criteria for Asian populations) are either not cited or not meaningfully integrated into the Discussion.
à We conducted an extensive literature search but were unable to identify recent large-scale cohort studies specifically examining the relationship between oral contraceptive use and metabolic syndrome in Asian populations that would provide directly comparable findings to our study. If the reviewer could direct us to specific critical modern cohort studies that we may have missed, we would be grateful to review and incorporate them into our Discussion.
We have meaningfully integrated the 2024 U.S. Medical Eligibility Criteria (US MEC) into our Discussion section, where we note that these guidelines recommend caution with combined oral contraceptives in women with existing cardiometabolic risk factors, particularly those with hypertension or diabetes, which aligns with our findings.